# Intestinal Permeability in Children with Functional Gastrointestinal Disorders: The Effects of Diet

**DOI:** 10.3390/nu14081578

**Published:** 2022-04-11

**Authors:** Valentina Giorgio, Gaia Margiotta, Giuseppe Stella, Federica Di Cicco, Chiara Leoni, Francesco Proli, Giuseppe Zampino, Antonio Gasbarrini, Roberta Onesimo

**Affiliations:** 1Department of Women’s, Children’s and Public Health Sciences, A. Gemelli University Hospital Foundation, IRCCS, 00168 Rome, Italy; valentina.giorgio@policlinicogemelli.it (V.G.); peppe.stella3@gmail.com (G.S.); federica.dicicco@gmail.com (F.D.C.); chiara.leoni@policlinicogemelli.it (C.L.); francesco.proli@guest.policlinicogemelli.it (F.P.); giuseppe.zampino@policlinicogemelli.it (G.Z.); roberta.onesimo@policlinicogemelli.it (R.O.); 2Department of Medical and Surgical Sciences, A. Gemelli University Hospital Foundation, IRCCS, 00168 Rome, Italy; antonio.gasbarrini@policlinicogemelli.it

**Keywords:** intestinal permeability, functional gastrointestinal disorders, children, diet, pediatric nutrition

## Abstract

Functional gastrointestinal disorders (FGIDs) are very common and life-impacting in children and young adults, covering 50% of pediatric gastroenterologist consultations. As it is known, FGIDs may be due to alterations in the gut–brain axis, dysbiosis and dysregulation of intestinal barrier, causing leaky gut. This may enhance increased antigen and bacterial passage through a damaged mucosa, worsening the impact of different medical conditions such as FGIDs. Little is known about the role of nutrients in modifying this “barrier disruption”. This narrative review aims to analyze the clinical evidence concerning diet and Intestinal Permeability (IP) in FGIDs in children. We searched the PubMed/Medline library for articles published between January 2000 and November 2021 including children aged 0–18 years old, using keywords related to the topic. Since diet induces changes in the intestinal barrier and microbiota, we aimed at clarifying how it is possible to modify IP in FGIDs by diet modulation, and how this can impact on gastrointestinal symptoms. We found that) is that small changes in eating habits, such as a low-FODMAP diet, an adequate intake of fiber and intestinal microbiota modulation by prebiotics and probiotics, seem to lead to big improvements in quality of life.

## 1. Introduction

Functional gastrointestinal disorders (FGIDs) are very common in children, since they represent 50% [1] of pediatric gastroenterologists’ consultations. These disorders are associated with a reduced quality of life, excess use of healthcare services, school absenteeism, anxiety and depression [2]. FGIDs are defined as a combination of clinical patterns characterized by chronic or recurrent gastrointestinal symptoms not explained by biochemical or structural alterations. These disorders are currently diagnosed using the Rome IV criteria, defined in 2016 [3]. They also have an effect on life in adulthood in terms of persistence of gastrointestinal symptoms, i.e., abdominal pain and discomfort, and psychological discomfort with an important impact on quality of life [4].

The pathogenesis of FGIDs, and in particular of Irritable Bowel Syndrome (IBS), which is the most common, remains unclear. Different mechanisms have been proposed, such as the presence of a “leaky-gut” with increased Intestinal Permeability (IP), visceral hypersensitivity, abnormal gut motility, small intestinal bacterial overgrowth, psychosocial factors and dysregulated gut–brain axis. Diet and nutrition seem to have a leading role; in fact, as stated by Yang, Q. et al., nutrients can interfere with gut microbiota, favoring the prevalence of some phyla over others and consequently modulating the functions of the intestinal barrier, therefore acting on gastrointestinal motility, sensitivity and barrier function [5].

It is known that there is a close connection between the gut and the brain, and that cross-communication occurs regularly. The central nervous system (CNS) controls the gut microbiome composition through peptides. Neuropeptides such as substance P, calcitonin gene-related peptide and neuropeptide Y (NPY) and vasoactive intestinal polypeptide (VIP) allow bidirectional gut–brain communication and can influence microbiota activity and its interaction with the brain [6]. Furthermore, the hypothalamic–pituitary–adrenal (HPA) axis releases cortisol—a crucial factor in depression and anxiety disorders—which regulates intestinal motility, integrity and hypersecretion [7]. Since communication is bidirectional, the intestinal microbiota can consecutively modulate the activity of the CNS through neural, endocrine, immune, and metabolic mechanisms [6].

The intestinal barrier plays an important role in this bidirectional interaction. This is a functional unit made up by the intestinal microbiota, the mucous layer, the intestinal epithelium, the components of natural and acquired immunity, the endocrine and neuroenteric system, the vascular–lymphatic system and digestive enzymes. It is the first line of defense against toxic, immunogenic and pro-inflammatory factors, placing in constant equilibrium the antigenic charge of the intestinal lumen with the complex organization of the intestinal mucosa [8]. It is essential for maintaining optimal health and preventing intestinal and systemic inflammation and immune activation. In fact, the normal intestinal barrier allows only small amounts of antigens to pass the mucosa to interact with the innate and adaptive immune systems. If this function is altered, it can lead to enhanced antigen and bacterial passage and, in turn, to pathological conditions [9]. The intestinal epithelium, composed of connected enterocytes, is one of the main constituents of intestinal barrier. It is formed of resistant, occlusive intracellular junctions (TJs). It is reinforced by a dense film of mucus and is in stable interaction with luminal contents and the enteric microbiota. In addition, short-chain fatty acids (SCFAs) are a significant basis of energy for enterocytes and are crucial signaling compounds for the preservation of gut health and its barrier function [10].

The intestinal epithelium regulates a “selective” gut permeability. Enterocytes are connected through intercellular junctions that include the following: (i) TJs, located more superficially, and (ii) adherent junctions and desmosomes, more deeply. TJs are made up of the extracellular binding domains (homo and heterotopic) of different types of proteins; they are very important in determining the selectivity of IP since they regulate and modulate paracellular absorption. They constitute transcellular pores by binding themselves with other transmembrane proteins called occludins, and with proteins of the junctional complex, such as zonula occludens (ZO)-1, (ZO)-2 and (ZO)-3, and Cytoskeletal proteins, such as microtubules and microfilaments [11]. TJs are continuously modulated and regulated by both intra- and extra-luminal signals. As an example, food and bacterial toxins act on ZO, causing an increase in absorption and IP. In vivo and in vitro studies have shown that bacteria can directly alter TJs through the release of soluble peptides, toxins, metabolites and bacterial membrane components [12,13].

Selective IP is one of the main qualities of the intestinal barrier, allowing molecules to pass through by non-mediated diffusion. As literature shows, the transport of molecules across the intestinal epithelium takes place via the transcellular and paracellular routes. Determining the urinary excretion of disaccharides and monosaccharides and their excretion ratios is a valid method for the measurement of IP. Monosaccharides, such as mannitol (M), pass through the transcellular routes, reflecting the degree of absorption of small molecules (defined as molecules with a diameter <0.65 nm). Disaccharides, such as lactulose (L), pass through the intercellular junction complex, reflecting the permeability of large molecules (defined as molecules with a diameter >0.93 nm). The L and M urinary excretion ratio (L/M ratio) is the most used test for the assessment of IP and eventually malabsorption [14]. An abnormal intestinal permeability—defined as a “leaky gut”—is one of the key pathological events for gastrointestinal diseases such as IBS and other FGIDs.

Nutritional support is a primary therapy for some gastrointestinal diseases, i.e., Crohn’s disease, as it allows the inflammatory activity to be controlled and may be an alternative to pharmacological treatment [15].

Little is known about the role of diet and nutrients on alterations of IP in pediatric patients with FGIDs. Given the effect that diet has on the intestinal microbiota, and given the known ability of microbiota to act on enterocytes and TJs and, indirectly, on IP, our study aimed at understanding the relationships between specific nutrients, host metabolism function, gut microbiota and IP. Our attempt was to further clarify the complex pathogenesis of FGIDs and investigate possible therapeutic strategies, in order to reduce gastrointestinal symptoms and improve quality of life.

## 2. Materials and Methods

### 2.1. Search Strategy

We performed a PubMed/Medline search for articles published between January 2000 and November 2021, using the search terms “intestinal permeability,” “children functional gastrointestinal disorders,” “diet and intestinal permeability” “gut microbiota and leaky gut”, “prebiotics and leaky gut” and the above MeSH keywords, with restriction for age < 18 years old, without restrictions on language or sex. We also searched manually for references of extracted articles. 

### 2.2. Selection Criteria

Five hundred and fifty-two papers, including one hundred and twenty reviews and meta-analyses, were screened. Out of them, 15 were included in the final analysis according to the selected criteria [16,17,18,19,20,21,22,23,24,25,26,27,28,29,30].

## 3. Diet and IP in FGIDs

While understanding the pathophysiology of FGIDs, the role of diet cannot be overlooked. During the process of digestion and absorption, intestinal epithelium and foods interact closely with different types and proportions of microorganisms in the GI tract. These interactions, through the generation of molecules such as neuropeptides and SCFAs, or favoring the proliferation of different bacterial phyla in the microbiota itself, have effects on the intestinal barrier, being able to modify its function causing pathologies or bringing benefits on GI health.

### 3.1. Fermentable Oligo-, Di-, Mono-Saccharides, and Polyols (FODMAPs)

Fermentable carbohydrates, including short-chain fermentable carbohydrates such as lactose, fructose, fructans, galactans, and polyalcohols, are small and osmotically active carbohydrates that are poorly absorbed in the small intestine and are rapidly fermented by the colonic microbiota. They have many healthy effects, such as promoting normal intestinal transit and inducing microbiota homeostasis; in addition, they appear to have effects on micronutrient absorption and the immune system [16]. FODMAPs are difficult to absorb and have been shown to contribute to symptoms in IBS and other FGIDs. Many mechanisms are involved, such as the increase in water volume in the small intestine, which leads to abdominal pain and bloating, bacterial fermentation, which produces colonic gases resulting in altered IP, luminal distension, and augmented intestinal motility [17]. Moreover, FODMAPs affect GI motility, reducing oro-cecal transit as demonstrated by gastrointestinal scintigraphy [18].

Consistent with this finding, Chumpitazi et al., in a randomized-controlled trial conducted on 33 children, demonstrated that a low FODMAP diet decreases IBS symptoms in children. In fact, after 2 days of low-FODMAP diet, children had fewer daily abdominal pain episodes, decreased median pain severity and lower breath hydrogen production [19]. The presumed mediator of this diet efficacy is decreased microbial fermentation, with benefits on intestinal barrier function and homeostasis. More, Staudacher HM found that a 4-week low-FODMAP diet decreases luminal *Bifidobacteria* versus a habitual diet, ameliorating dysbiosis and leaky gut [20].

### 3.2. Dietary Fiber

Dietary fibers are defined as those organic substances that our digestive system enzymes are not able to break down and, therefore, digest. They are a group of carbohydrate polymers, oligomers, and lignin that escape digestion in the small intestine and reach the colon intact, where they are partially or completely fermented by the gut microbiota [21]. Dietary fiber contributes to fecal bulking, directly via their own mass and/or by the mass of the water that they attract, and indirectly by stimulating the growth of colonic microbiota, leading to an increase in microbial biomass. They are categorized as soluble or insoluble. Soluble fibers are rich in oatmeal, nuts, seeds, beans, lentils, peas, and some fruits such as apples, blueberries, and vegetables. They are fermented in the colon and absorb water, forming a gel and making the stool softer and easier to pass. Insoluble fibers are subject to limited fermentation and have a bulking action. They are found in wheat bran, vegetables, and whole grains but are not directly effective for relieving symptoms. At least twenty-six different types of fibers are known. Out of all types of fiber, guar gum, glucomannan, cocoa husk, psyllium, inulin, corn fiber, acacia fiber, oligosaccharides, and soy fiber have been studied for the treatment FGIDs in children. One of the main features, as seen before, is that they can influence microbiota composition. In particular, dietary fibers are fermented by enteric bacteria in the colon to produce short-chain fatty acids (SCFAs) such as acetate, propionate, and butyrate. SCFAs and other metabolites mediate beneficial effects, including the preservation of energy homeostasis, and inflammatory signal inhibition and immunity and, above all, gut epithelial integrity and permeability [22]. Misconceptions about the etiology and management of FGIDs are common: a low-fiber diet has been indicated as the most common cause of childhood constipation by 32% of primary care physicians in three Western countries [23], even if it has been reported that constipated children may have a lower, equivalent or higher intake of fiber [24]. As demonstrated by Axelrod CH et al. [25], pediatric studies on the use of dietary fiber for FGIDs treatment have shown its benefit in reducing pain episodes, improving stool frequency and consistency and reducing bowel movement frequency. However, not all studies are in line with what previously said; in fact, Shulman et al., in a study on the effects of Psyllium fiber on intestinal permeability in children with IBS, showed that this fiber reduced the number of abdominal pain episodes but did not alter gut permeability or microbiome composition [26]. In conclusion, literature shows that children consume a suboptimal amount of dietary fiber, due to the limited intake of vegetables, fruits, and whole grains. The actual literature evidence suggests to prefer a fiber intake not higher than that adequate one for age (age + 3 to 5 g/day). It is therefore necessary to favor the correct intake of common sources of soluble fiber such as fruit, vegetables, oatmeal, barley, beans, lentils, and peas, rather than taking supplements. More studies are needed to characterize and understand the correct use and dosage of dietary fibers in the pediatric age in order to modulate GI symptoms in FGIDs.

### 3.3. Prebiotics

Prebiotics are referred to as “Selectively fermented ingredients that result in specific changes in the composition and/or activity of the gastrointestinal microbiota, thus conferring benefit(s) upon host health” [27]. Most prebiotics are nondigestible oligosaccharides. The vast majority of studies have focused on inulin, fructooligosaccharides (FOS), and galactooligosaccharides (GOS). Our diet comprises several foods that contain FOS and inulin, such as onion, garlic, banana, chicory, wheat, and some cereals. On the other hand, GOS have a dairy origin and result from the enzymatic conversion of lactose using beta-galactosidase. One of the main effects is that prebiotics promote intestinal barrier function through the modulation of intestinal TJs. Prebiotics’ direct effects on the gut microbiota is a reasonable explanation for these changes in TJ protein expression and distribution. As demonstrated by Chang B and co-authors, prebiotics can interact with microbiota, preventing the passage of endotoxins and other bacterial products from the intestinal lumen into the circulation. This may lead to a down-regulation in the expression of TNFα, which down-regulates TJ expression and increases epithelial permeability [28]. Given that prebiotics stimulate the growth of select probiotic species, it follows that prebiotic supplementation mirrors the beneficial effects produced by probiotics themselves [29]. According to this, Wegh CAM et al. showed that inulin, FOS, and GOS, even when consumed in small amounts (0.24–0.8 g/100 mL formula in infants or 1.5–5 g/day in young children), promote the growth of *Bifidobacterium* and *Lactobacillus* species [30].

## 4. Role of Microbiota

The human intestine hosts trillions of microbial cells in a symbiotic relationship with the host and plays a vital role in health and disease. The microbiota is composed of different bacteria species taxonomically classified by species, genus, family, order, class, and phylum. The phyla *Firmicutes* and *Bacteroidetes* represent 90% of gut microbiota [7]. While the specific microbial composition varies among individuals, the functional role is conserved, providing essential nutrients, metabolizing dietary fiber into short-chain fatty acids, and ensuring the proper development of the immune system [31]. Dysbiosis provokes the disruption of the intestinal barrier tight junctions (TJs), leading to increased intestinal permeability [32,33]. Confirming this, dysbiosis has been reported as a pathophysiologic factor in FGIDs [8]. Multiple studies have demonstrated differences in the fecal microbial communities of patients with IBS compared to healthy controls. Patients with IBS show reductions in the relative abundance of *Bifidobacterium* and *Lactobacillus*, *Collinsella aerofaciens*, *Coprococcus eutactus*, *and Clostridium cocleatum*, and an increase in *Enterobacteriaceae* and in the *Firmicutes:Bacteroidetes* ratio [34]. Furthermore, *Haemophilus*, *Dorea*, and *Veillonella* were more abundant with reduced potential butyrate-producing *Eubacterium* and *Anaerovarax* [35]. Moreover, microbiota composition seems to correlate with abdominal pain severity and frequency and could be used to distinguish IBS subtypes. Fecal microbial community composition also might be used to predict children who are more likely to respond to a low-FODMAP diet [36].

### Probiotics

To further elucidate the role of the intestinal microbiota in FGIDs, multiple studies have examined the effects of probiotic supplementation. As defined by the WHO, probiotics are live microorganisms that, when administered in adequate amounts, confer a health benefit to the host. They are used to manage dysbiosis, restore the microbe diversity, and reestablish disturbed gut microbiota. The most-studied probiotic in pediatric FGIDs is *Lactobacillus rhamnosus GG* [37]. Francavilla et al. [38] compared *Lactobacillus GG* versus placebo in 83 patients with IBS and 58 patients with FAP. They found that after a 4-week run-in and 8-week intervention, both pain intensity and frequency were significantly lower in children with probiotics than those with the placebo. Moreover, they demonstrated that children treated with LGG had an improvement in intestinal permeability compared to the pre-treatment test. This is a known property of LGG, as it is known its role in preventing the breakdown of tight junctions caused by *E.*
*coli* [39] and allowing the secretion of proteins that stabilize tight junctions [40] through the stimulation of Toll-like receptor-2 by a ligand-mediated stimulation, as well as through the modulation of the expression of constituents of TJs [41], in particular of ZO-2, occludin and several claudins [42].

## 5. Discussion

As literature shows, FGIDs are some of the most life-impacting diseases of school-age children, adolescents and young adults, both in physical and psychological terms, with an important impact on quality of life. Over the years, with an increasing number of patients affected by FGIDs, our clinical knowledge has increased, but we still know little about the pathophysiology and the most appropriate therapy. We have wide evidence of the leading role played by the gut–brain axis and its intersection with the intestinal barrier and microbiota.

These three elements work together to provide essential nutrients, metabolize dietary fiber into SCFAs, and ensure the proper development of the immune system through the regulated passage of small quantities of antigens. CNS controls the gut microbiome composition through peptides, and vice versa, all through the intestinal barrier.

A dysregulation of this system leads to a leaky gut, which seems to be one of the main factors responsible for FGIDs, in the form of increased intestinal permeability and transit, abdominal pain, the altered metabolism of nutrients and, consequently, an increase in bloating and abdominal discomfort. All this can have psychological repercussions, activating the HPA axis, which in turn acts on the gut, triggering a vicious circle.

Many studies have been conducted in order to understand which is the best therapy for patients with FGIDs, concentrating on single therapeutic possibilities, each of which has shown that the intestinal barrier plays a leading role. The interventions performed are various and the main results and outcomes of the evaluated RCTs are reported in Table 1.

Starting from the nutritional side, as shown in Figure 1, it has been demonstrated that a low-FODMAP diet, reducing carbohydrate fermentation and modulating microbiota, improves intestinal permeability, and so, intestinal discomfort in terms of bloating, bowel movements and number and peaks of abdominal pain episodes. There is still little clarity regarding the role played by dietary fibers, being able to bring benefits, as stated by Axelrod CH et al., or a symptom’s worsening as demonstrated by Shulman et al., but the major part of the studies currently available seem to demonstrate their benefit if administered in adequate quantities for age, and possibly from natural sources. These positive effects can be explained because dietary fibers are fermented by microbiota in SCFAs that mediate gut epithelial integrity and improve IP.

We must not forget about prebiotics: those are selectively fermented ingredients, which promote intestinal barrier function through the modulation of intestinal TJs. Prebiotics work by cross talking with microbiota, improving its composition and/or activity conferring benefit(s) upon host health.

As for Intestinal Microbiota and Probiotics, which are strictly associated, they have an important role in modulating IP. It is now known how the microbiota of subjects with FGIDs differs from that of healthy ones, with the lack of *Bifidobacterium* and *Lactobacillus* and an increase in *Enterobacteriaceae* and in the *Firmicutes:Bacteroidetes* ratio. Even more, different FGIDs have different characteristics in the microbiota. The probiotic that, most of all, has beneficial effects is the *Lactobacillus GG*. Numerous studies have shown how this important symbiont can stabilize the intestinal barrier, in particular the TJs, reducing intestinal permeability and, consequently, improving physical and phycological symptoms in patients with FGIDs.

All of these nutritional approaches can have benefits on the general well-being of the child and their family. A diet that favors physical health, reducing abdominal bloating and pain episodes and improving stool consistency, also has repercussions on the psychological well-being and quality of life of the child. A healthy child is one of the fundamental building blocks for the well-being of the whole family. Providing specific dietary recommendations and nutritional support to families with children with FGIDs is an important medical challenge to be achieved to comprehensively treat these disorders.

## 6. Conclusions

The intestinal barrier plays a key role in improving or worsening symptoms in functional gastrointestinal disorders. Small changes in eating habits seem to lead to big improvements in quality of life. Many nutritional aspects - such as low FODMAPs, adequate intake of fiber, and intestinal microbiota modulated by prebiotics and probiotics - have an important positive effect on intestinal permeability. Our knowledge about the complex communication mechanism of the gut–microbiota–brain axis is still scarce, and further studies are needed to better define these aspects.

## Figures and Tables

**Figure 1 nutrients-14-01578-f001:**
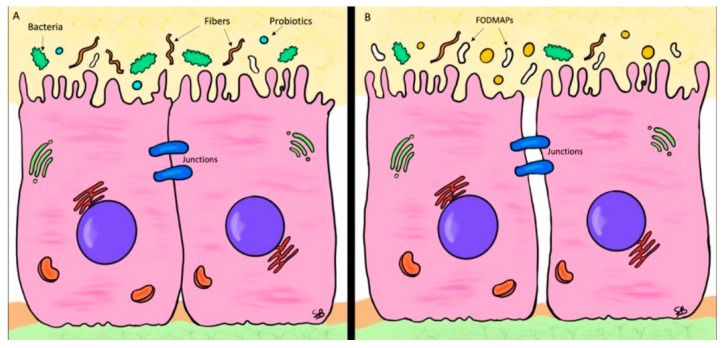
Normal vs. leaky gut (**A**) Normal enterocytes characteristic of a healthy gut. Thanks to the barrier balance, due to microbiota, fibers and pre- and pro-biotics, the TJs are intact and IP is preserved; (**B**) Typical epithelium of a leaky gut. TJs are malfunctioning due to an excess of FODMAPs, dysbiosis and scarcity of fibers and pre- and pro- biotics.

**Table 1 nutrients-14-01578-t001:** Clinical trials included in the review, main methods, results and outcomes.

First Author and Year of Pubbication	Study Design	Cases (n.)	Age ± SD (ys)	FGIDs	Intervention	Clinical Outcome	Other Outcomes	Results	Other Results
Shulman et al. 2017 [26]	RCT	103	13.0 ± 3.0	IBS	Use of psyllium fiber	Abdominal pain and stool patterns	Breath hydrogen or methane production, gut permeability, microbiome composition	Reduction in the number of pain episodes, no differences in stool patterns	Breath hydrogen or methane production, intestinal permeability and microbiome composition were similar between groups.
Chumpitazi et al. 2015 [19]	RCT	52	12.0 ± 5.0	IBS	Low FODMAPs diet	Daily abdominal pain episodes	Microbiome composition	Reduction in the mean number of pain episodes	Gut microbiome biomarkers may be associated with low FODMAP diet efficacy
Chumpitazi et al. 2014 [36]	Prospective	12	10.9 ± 3.6	IBS	Low FODMAPs diet	Abdominal pain frequency	Potential microbial factors related to diet efficacy, breath hydrogen and methane, whole intestinal transit time	Decreased GI symptoms	Microbial factors such as gut microbiome composition and stool metabolites related to low FODMAPs diet efficacy
Francavilla et al. 2010 [38]	RCT	141	6.5 ± 2.1	IBS	Treatment course with Lactobacillus Rhamnosus GG (LGG)	Abdominal pain at the end of the intervention period	Differences in IP	Significant reduction of frequency and severity of abdominal pain	Significant improvement- in terms of reduction- of IP

Abbreviation: RCT randomized controlled trial, IBS irritable bowel syndrome, FODMAPs fermentable olygo-di-mono-saccharides and polyols, IP intestinal permeability, GI gastrointestinal.

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
