# Peer review of "Intestinal Permeability in Children with Functional Gastrointestinal Disorders: The Effects of Diet"

_nutrients, 2022, doi:10.3390/nu14081578_

Round 1

Reviewer 1 Report

The authors present an interesting review which evaluates the various dietary strategies utilised to modulate intestinal permeability in pediatric patients with functional gastrointestinal disorders. While the paper is relevant, in order to elucidate the importance/significance of the subject matter an expansion of the text, and an explanation of mechanisms, would be highly beneficial.

Abstract: Please expand this section as it does not provide an adequate description of the body of work undertaken.

Line 14: “this review... testinal Disorders in children” The sentence is incomplete?? Missing words as it does not link to first few words on line 15.

What do the authors mean by “…… and in molecules production?” Please clarify Line 15-16

Line 17-18 “main dietary strategies associated with lower generation of inflammation are reported”. Please tie this in with the abstract introduction. No mention of inflammation but of course alteration of barrier function is associated with pro-inflammatory status within the intestinal lumen.

Description of methods is lacking in the abstract.

Introduction: Please structure such that the anatomy and physiology of the intestine is briefly explained, which will then set the scene for explaining diffusive mechanisms incl. transcellular vs paracellular transport which are key elements that account for intestinal permeability (IP), and hen the manner in which IP is measured. Again throughout the Introduction inflammation/inflammatory processes are not addressed. References are not adequately cited. What impact does IP or FGIDs during childhood have on later life? Why is it important? 

Line 24: Please provide a ref for the first statement. 50% is a specific number. 

Line 35: what do the authors mean by nutrients can interfere with gut microbiota? Please clarify

Line 39: how does the CNS control gut microbiome composition through peptides? What are the authors specifically referring to? 

Line 45: Please clarify and explain what is meant by the antigenic charge of the intestinal lumen

Materials and Methods

Line 77: ."..articles between 2000 and November 2021". Month missing. 

Line 82: Selection Criteria - please provide a CONSORT flowchart/diagram detailing articles included in the review

Line 84 "....30 were included in the final analysis" WHat was the final analysis conducted??? Please describe. Have the authors undertaken a systematic review? 

Results: Please provide a Table of all studies with description of subjects, duration of study, how was IP tested?, intervention with dose etc. 

In the current format the implications of diet vs its role in FGID /IP is not apparent. 

Line 142: (age +3 to 5 g/die). Do the authors mean day?? 

Line 144: "more studies are needed" Please elaborate/expand.... what the specific purpose of these studies aught to be. Presently unclear. 

Line 154-155: "Prebiotics direct effects on the gut microbiota is a reasonable explanation for these changes in TJs protein expression and distribution." Please clarify and explain what the authors mean by this? 

Line 189: scientifically would not use the term significantly smaller. Please consider revising to significantly lower? 

Line 193-194: how do lactobacillus GG 'stabilize' tight junctions?  Clarify. 

DISCUSSION: Please expand to provide a holistic approach to the review. Presently does not provide a detailed discussion of the literature reviewed. Adequate additional references are required. 

Other comments:

Standardise acronyms and proof read document for typos and use of capital letters mid-sentence.

Figure 1 is not cited in text. Recommended to improve the figure to capture overall understanding of the review - presently is only related to FODMAPS. 

Reviewer 2 Report

This brief review brings together knowledge on the effects of selected food components on intestinal permeability in children with functional gastrointestinal disorders from nearly 21 years of worldwide research, searched PubMed/Medline. While the text seems to be a good concise summary, the explanation of why these specific selected factors were chosen is lacking.

Minor comments:

  1. The abstract section needs to be corrected because some text is missing between lines 14 and 15.
  2. The authors should clearly state the hypothesis of this review study in the introductory section. They must have expected that particular factors, food components should have a large effect on intestinal barrier permeability.
  3. Check spelling in the text, such as in line 63.
  4. The starting point for article searches is missing - what month of the year 2000 is it? 2021 was not whole so was the start point January 2000?
  5. Line 142 – what does “die” mean?
